# *Rhodoalgimonas zhirmunskyi* gen. nov., sp. nov., a Marine Alphaproteobacterium Isolated from the Pacific Red Alga *Ahnfeltia tobuchiensis*: Phenotypic Characterization and Pan-Genome Analysis

**DOI:** 10.3390/microorganisms11102463

**Published:** 2023-09-30

**Authors:** Olga Nedashkovskaya, Nadezhda Otstavnykh, Larissa Balabanova, Evgenia Bystritskaya, Song-Gun Kim, Natalia Zhukova, Liudmila Tekutyeva, Marina Isaeva

**Affiliations:** 1G.B. Elyakov Pacific Institute of Bioorganic Chemistry, Far Eastern Branch, Russian Academy of Sciences, Prospect 100 Let Vladivostoku, 159, Vladivostok 690022, Russia; chernysheva.nadezhda@gmail.com (N.O.); lbalabanova1@gmail.com (L.B.); belyana@gmail.com (E.B.); 2Korean Collection for Type Cultures, Biological Resource Center, Korea Research Institute of Bioscience and Biotechnology, 181 Ipsin-gil, Jeongeup-si 56212, Republic of Korea; sgkim@kribb.re.kr; 3A.V. Zhirmunsky National Scientific Center of Marine Biology, Far Eastern Branch, Russian Academy of Sciences, Palchevskogo Street 17, Vladivostok 690041, Russia; nzhukova35@list.ru; 4Innovative Technology Center, Far Eastern Federal University, 8 Suhanova St., Vladivostok 690950, Russia; tekuteva.la@dvfu.ru; 5ARNIKA, Territory of PDA Nadezhdinskaya, Centralnaya St. 42, Volno-Nadezhdinskoye, Vladivostok 692481, Russia

**Keywords:** marine bacteria *Rhodoalgimonas zhirmunskyi*, *Roseobacteraceae*, red alga *Ahnfeltia tobuchiensis*, whole genome sequence, pan-genome analysis

## Abstract

A novel Gram-staining negative, strictly aerobic, rod-shaped, and non-motile bacterium, designated strain 10Alg 79^T^, was isolated from the red alga *Ahnfeltia tobuchiensis*. A phylogenetic analysis based on 16S rRNA gene sequences placed the novel strain within the family *Roseobacteraceae*, class *Alphaproteobacteria*, phylum *Pseudomonadota*, where the nearest neighbor was *Shimia sediminis* ZQ172^T^ (97.33% of identity). However, a phylogenomic study clearly showed that strain 10Alg 79^T^ forms a distinct evolutionary lineage at the genus level within the family *Roseobacteraceae* combining with strains *Aquicoccus porphyridii* L1 8-17^T^, *Marimonas arenosa* KCTC 52189^T^, and *Lentibacter algarum* DSM 24677^T^. The ANI, AAI, and dDDH values between them were 75.63–78.15%, 67.41–73.08%, and 18.8–19.8%, respectively. The genome comprises 3,754,741 bp with a DNA GC content of 62.1 mol%. The prevalent fatty acids of strain 10Alg 79^T^ were C_18:1 ω7c_ and C_16:0_. The polar lipid profile consisted of phosphatidylethanolamine, phosphatidylglycerol, phosphatidylcholine, an unidentified aminolipid, an unidentified phospholipid and an unidentified lipid. A pan-genome analysis showed that the unique part of the 10Alg 79^T^ genome consists of 13 genus-specific clusters and 413 singletons. The annotated singletons were more often related to transport protein systems, transcriptional regulators, and enzymes. A functional annotation of the draft genome sequence revealed that this bacterium could be a source of a new phosphorylase, which may be used for phosphoglycoside synthesis. A combination of the genotypic and phenotypic data showed that the bacterial isolate represents a novel species and a novel genus, for which the name *Rhodoalgimonas zhirmunskyi* gen. nov., sp. nov. is proposed. The type strain is 10Alg 79^T^ (=KCTC 72611^T^ = KMM 6723^T^).

## 1. Introduction

Red algae are widely distributed in marine environments and tend to be colonized by rich and diverse communities of bacteria, some of which may be cultured under laboratory conditions [1,2,3,4]. Many novel bacterial species associated with different red algae have been described in recent years. The strains belonging to a new genus and species *Algibacillus agarilyticus* (the family *Alteromonadaceae*, class *Gammaproteobacteria*), and two new species *Marinomonas algarum* (the family *Oceanospirillaceae*, class *Gammaproteobacteria*) and *Maribacter algarum* (the family *Flavobacteriaceae*, class *Flavobacteriia*) were isolated from the red alga *Gelidium amansii* collected from coastal waters in Wenhai (Shandong province, China) [5,6,7]. The novel members of the genera *Marinomonas* and *Maribacter*, *Marinomonas agarivorans* and *Maribacter algicola*, were also recovered from the surfaces of the red algae *Gracillaria bladgettii* and *Porphyridium marinum*, respectively [8,9]. The representatives of three new species of the family *Flavobacteriaceae*, *Muricauda* (formerly *Flagellimonas*) *algicola*, *Psychroserpens luteolus,* and *Tenacibaculum aquimarinum,* were described as epiphytic bacteria of the red algae *Asparagopsis taxiformis*, *Gelidium* sp., and *Chondrus* sp. collected from Korean and Chinese coastal waters [10,11,12]. The new species *Thalassotalea algicola* (family *Colwelliaceae*, class *Gammaproteobacteria*) was isolated from a red alga *Porphyra* sp. collected from the Chinese coast in Weihai [13]. The new genus and species *Hwanghaeella grinnelliae* (the family *Rhodospirillaceae*, class *Alphaproteobacteria*) was isolated from a red marine alga *Grinnellia* sp. in the Yellow Sea of the Republic of Korea [14]. Other representatives of the class *Alphaproteobacteria* associated with the Pacific red alga *Polysiphonia* sp. were placed as new species of the family *Roseobacteraceae* (formerly *Rhodobacteraceae*), *Roseibium* (formerly *Labrenzia*) *polysiphoniae* [15]. The new species *Aquimarina algiphila* (the family *Flavobacteriaceae*, class *Flavobacteriia*), and the new genus and species *Algicella marina* (the family *Paracoccaceae*, class *Alphaproteobacteria*) were recovered from a common inhabitant of the Sea of Japan, *Tichocarpus crinitus* [16,17]. Three new species of the family *Flavobacteriaceae*, *Aureibaculum algae*, *Flavobacterium ahnfeltiae,* and *Polaribacter staleyi*, were isolated from the red alga *Ahnfeltia tobuchiensis* collected from coastal waters of the Okhotsk Sea [17,18,19].

The aim of this report is to provide a genomic characterization and taxonomic description of another associate of the red alga *Ahnfeltia tobuchiensis*, designated as strain 10Alg 79^T^ (KMM 6723^T^). This isolate is proposed as a new member of the family *Roseobacteraceae* (formerly *Rhodobacteraceae*) based on the study of its phenotypic, chemotaxonomic, and genomic characteristics.

## 2. Materials and Methods

### 2.1. Bacterial Isolation and Maintenance

Strain 10Alg 79^T^ was isolated from the red alga *Ahnfeltia tobuchiensis* collected near Island Paramushir, Kuril Isls, Okhotsk Sea, Russia, using a standard dilution plating method. The sample of algal fronds (5 g) was homogenized in 10 mL sterile seawater in a glass homogenizer, and 0.1 mL of homogenate was spread onto Marine Agar 2216 (MA, Difco) plates. The novel isolates were obtained from a single colony after incubation of the plate at 28 °C for 7 days. After primary isolation and purification, the strains were cultivated at 28 °C on the same medium and stored at −80 °C in marine broth (Difco, Sparks, MD, USA) supplemented with 20% (*v*/*v*) glycerol. The type strains *Aquicoccus porphyridii* JCM 31543^T^ and *Marimonas arenosa* KCTC 52189^T^ obtained from the Japan Collection of Microorganisms (JCM) and the Korean Collection for Type Cultures (KCTC), respectively, were used as reference strains in the parallel tests for this study. The data for *Lentibacter algarum* ZXM100^T^ neighboring to the new isolate on the genomic tree were also included in all tables as a reference strain.

### 2.2. Phenotypic and Chemotaxonomic Characterization

The physiological, morphological, and biochemical characteristics of strain 10Alg 79^T^ were determined following standard methods. The novel isolate was also examined using the API 20E, API 20NE, API ID 32GN, API 50CH, and API ZYM galleries (bioMérieux, Marcy l’Etoile, France) according to the manufacturer’s instructions. All galleries were incubated at 28 °C. Cell morphology was examined using transmission electron microscopy using cells grown for 48 h on MA at 28 °C. Gram-staining was performed as recommended by Gerhardt et al. [20]. Oxidative or fermentative utilization of glucose was determined on Hugh and Leifson’s medium modified for marine bacteria [21]. Catalase activity was tested with the addition of 3% (*v*/*v*) H_2_O_2_ solution to a bacterial colony and observation for the appearance of gas. Oxidase activity was determined using N,N,N,N-tetramethyl-p-phenylenediamine. Degradation of agar, starch, casein, gelatin, chitin, DNA, and urea; growth at the pH 5–11 range (using increments of 1 pH unit); production of acid from carbohydrates, hydrolysis of Tweens 20, 40, and 80, nitrate reduction; and production of hydrogen sulfide were tested according to Smibert and Krieg [22]. The temperature range for growth was 4–42 °C for 1 °C intervals and was assessed on MA. Tolerance to NaCl was checked in a medium containing 5 g Bacto Peptone (Difco), 2 g Bacto yeast extract (Difco), 1 g glucose, 0.02 g KH_2_PO_4_, and 0.05 g MgSO_4_·7H_2_O per liter of distilled water with 0, 0.5, 1.0, 1.5, 2.0, 2.5, 3, 4, 5, 6, 8, and 10% (*w*/*v*) of NaCl. Susceptibility to antibiotics was examined using the routine disc diffusion plate method. Discs were impregnated with the following antibiotics: ampicillin (10 μg), benzylpenicillin (10 U), carbenicillin (100 μg), cefalexin (30 μg), cefazolin (30 μg), chloramphenicol (30 μg), erythromycin (15 μg), doxycycline (10 μg), gentamicin (10 μg), kanamycin (30 μg), lincomycin (15 μg), nalidixic acid (30 μg), neomycin (30 μg), ofloxacin (5 μg), olean domycin (15 μg), oxacillin (10 μg), polymyxin B (300 U), rifampicin (5 μg), streptomycin (30 μg), tetracycline (5 μg), and vancomycin (30 μg).

Fatty acid methyl esters and polar lipids of strain 10Alg 79^T^ and its closest phylogenetic relatives, *Aquicoccus porphyridii* JCM 31543^T^ and *Marimonas arenosa* KCTC 52189^T^, were extracted and analyzed as described previously [23] using cells grown on MA for 48 h at 28 °C. Isoprenoid quinones were extracted with chloroform/methanol (2:1, *v*/*v*) and purified with TLC using a mixture of n-hexane and diethyl ether (85:15, *v*/*v*) as the solvent. Isoprenoid quinone composition was characterized with HPLC (Shimadzu LC-10A, Shimadzu, Kyoto, Japan) using a reversed phase type Supelcosil LC-18 column (15 cm × 4.6 mm) and acetonitrile/2-propanol (65:35, *v*/*v*) as a mobile phase at a flow rate of 0.5 mL min^−1^, as described previously [24]. The column was kept at 40 °C. Quinones were detected by monitoring at 275 nm.

### 2.3. 16S rRNA and RpoC Sequences and Phylogenetic Analysis

Genomic DNA extracted from strain 10Alg 79^T^ with the NucleoSpin Tissue kit (Macherey-Nagel, Düren, Germany) was used to amplify 16S rRNA genes with 27F (5′-AGAGTTTGATCMTGGCTCAG-3′) and 1492R (5′-TACGGTTACCTTGTTACGACTT-3′) primers [25] and sequences using an ABI Prism 3130xL DNA analyzer (Applied Biosystems, Hitachi, Japan).

The analysis of 16S rRNA gene sequences was performed on the EzBioCloud server [26], and the phylogenetic analysis was conducted using MEGA X software, version 10.2.1 [27], with the neighbor-joining (NJ), maximum-likelihood (ML), and maximum-parsimony (MP) methods. Genetic distances were calculated according to the Kimura two-parameter model [28], and bootstrap values were obtained from 500–1000 alternative trees.

The maximum-likelihood phylogeny of RpoC sequences translated from genome sequences was calculated using the IQ-TREE web server, version 1.6.12 [29], with 100 non-parametric bootstrap replicates and the LG+I+G4 substitution model determined using ModelFinder [30].

### 2.4. Whole-Genome Sequencing and Genome-Based Phylogenetic Analysis

The genomic DNA of strain 10Alg 79^T^ extracted as described above in Section 2.4 was verified using DNA gel electrophoresis and quantified in a Qubit 4.0 Fluorometer (Thermo Fisher Scientific, Waltham, MA, USA). The DNA sequencing library was prepared using Nextera DNA Flex kits (Illumina, San Diego, CA, USA) and was sequenced using paired-end runs on an Illumina MiSeq platform with a 150 bp read length. The row reads were trimmed using Trimmomatic, version 0.39 [31], and their quality was assessed using FastQC, version 0.11.8 (https://www.bioinformatics.babraham.ac.uk/projects/fastqc/, accessed on 21 August 2021). Filtered reads were assembled into contigs with SPAdes, version 3.15.3 [32]. The quality of assembly was analyzed using QUAST, version 5.0.2 [33]. Genome completeness and contamination were estimated using CheckM, version 1.1.3, based on the taxonomic-specific workflow (family *Rhodobacteriaceae*) [34]. The genome was annotated using the NCBI Prokaryotic Genome Annotation Pipeline (PGAP), Rapid Annotation using Subsystem Technology (RAST), and the Pathosystems Resource Integration Center (PATRIC) [35,36,37]. Comparisons of the average nucleotide identity (ANI), average amino acid identity (AAI), and in silico DNA-DNA hybridization (dDDH) values of strain 10Alg 79^T^ and its closest neighbors were performed with the online server ANI/AAI-Matrix [38] and TYGS platform [39], respectively.

The phylogenomic analysis was performed using PhyloPhlAn software, version 3.0.1, based on a set of 400 conserved bacterial protein sequences [40]. A genome-wide analysis of orthologous clusters and singleton genes was carried out using OrthoVenn2 and OrthoVenn3 (https://orthovenn3.bioinfotoolkits.net/home, accessed on 15 August 2023) [41,42].

To identify carbohydrate-active enzymes (CAZymes), the dbCAN2 meta server, version 11, was used with default settings (http://cys.bios.niu.edu/dbCAN2, accessed on 1 December 2022) [43]. Predictions using two of the three algorithms integrated within the server (DIAMOND, HMMER, and dbCAN-sub) were considered sufficient for CAZy family assignments. The relative abundances of CAZymess were visualized with heat maps using pheatmap, version 1.0.12, in RStudio, version 2022.02.0+443, with R, version 4.1.3. Annotation of secondary metabolite biosynthetic gene clusters was conducted using antiSMASH server, version 7.0.0beta1-67b538a9 (https://antismash.secondarymetabolites.org/#!/start, accessed on 12 December 2022) [44]. The genomic regions containing NRPS-like fragment and Type I PKS (Polyketide synthase) biosynthetic gene clusters were extracted from GBK files of the genomes using Geneious Pro software, version 4.8 [45]. Generated GBK files were modified by adding custom color feature qualifiers, according to antiSMASH conventional coloring. Pairwise comparisons of each locus between four genomes were carried out using BLASTn (BLAST version 2.11.0+) run in EasyFig (version 2.2.5) [46]. Synteny plots were visualized using Easyfig with a minimum of 100 bp BLAST hits. Fonts and sizes in all figures were edited manually in Adobe Photoshop CC 2018 for better visualization.

The draft genome of *M. arenosa* KCTC 52189^T^ was obtained in this study as described above for the draft genome of 10Alg 79^T^ and used in the comparative genome analysis.

## 3. Results and Discussion

### 3.1. Phylogenetic and Phylogenomic Analyses

At first, the 1470 bp length 16S rRNA gene sequence of 10Alg 79^T^ was amplified and used to perform 16S rRNA comparisons on the EzBioCloud server [26]. From this analysis, 10Alg 79^T^ showed the highest similarity of 97.40% with ‘*Tropicibacter alexandrii*’ LMIT003^T^, followed by *Shimia sediminis* ZQ172^T^ (97.33%), *Shimia aestuarii* DSM 15283^T^ (96.75%), and *Aliishimia ponticola* MYP11^T^ (96.75%). However, the 16S rRNA phylogenetic trees reconstructed with ML (Figure 1); MP, and NJ algorithms demonstrated very low-support branches and topological discordance.

Therefore, the position of strain 10Alg 79^T^ was further determined using *rpoC* gene sequences extracted from type strain genomes of genera affiliated with the family *Roseobacteraceae* (formerly *Rhodobacteraceae*). RpoC protein sequences have been shown to be suitable for the *Roseobacter* group phylogeny [17,47]. An ML phylogenetic tree inferred on 43 RpoC protein sequences showed that 10Alg 79^T^ forms a poorly supported clade together with *M. arenosa* KCTC 52189^T^ [48], *A. porphyridii* L1 8-17^T^ [49], and *L. algarum* DSM 24677^T^ [50] (Figure 2).

Further, genomes of 18 type species of RpoC-related genera were selected for genomic tree building using PhyloPhlAn 3.0 software [40] using the concatenated sequence alignment of 400 conservative proteins. The resulting tree showed that 10Alg 79^T^ forms a distinct genus-level lineage (Figure 3). Despite the higher identity of 16S rRNA sequences with those of ‘*T. alexandrii*’ LMIT003^T^ (97.40%) and *S. sediminis* ZQ172^T^ (97.33%), 10Alg 79^T^ comprises the same clade with *M. arenosa* KCTC 52189^T^ (96.03%), *A. porphyridii* L1 8-17^T^ (96.17%), and *L. algarum* DSM 24677^T^ (95.95%) on the genomic tree.

The ANI/AAI values between the genomes of strains 10Alg 79^T^ and strains of three closely related genera *M. arenosa* KCTC 52189^T^, *A. porphyridii* L1 8-17^T^, and *L. algarum* DSM 24677^T^ were 78.15%/73.08%, 77.85%/71.35%, and 75.63%/67.41%, respectively. In addition, the 10Alg 79^T^ genome shared low values of dDDH (formula d4) with *M. arenosa* KCTC 52189^T^ (19.4%), *A. porphyridii* L1 8-17^T^ (19.8%), and *L. algarum* DSM 24677^T^ (18.8%).

These values were significantly below the boundaries for the demarcation of bacterial species [51,52]. The AAI values between 10Alg 79^T^ and the representatives were 67.41–73.08%, which did not exceed the declared genus boundaries of 80% [53]. Thus, the phylogenomic analysis based on the genome relatedness indexes clearly supported 10Alg 79^T^ as a novel type species of a new genus in the family *Rosebacteraceae* (formerly *Rhodobacteraceae*).

Remarkably, a search for 16S rRNA sequences homologous to 10Alg 79^T^ in NCBI, Greengenes, EzBioCloud, and Silva databases revealed only two 16S rRNAs with 99% identity (JQ215453.1 and JQ212029.1). Both originated from uncultured bacterium clones isolated from the dolphin *Tursiops truncates* in 2011 and 2013. This means that 10Alg 79^T^ represents a new species of the rare genus of the family *Roseobacteraceae*.

### 3.2. Genomic Characteristics and Analysis of Genus-Related Features

The draft genome of strain 10Alg 79^T^ was de novo assembled into 73 contigs, with an N_50_ value of 702,161 bp and an L_50_ value of two. The genome size was estimated at 3,754,741 bp in length with an overall GC content of 62.1%. The genome-extracted 16S rRNA gene sequence was 100% identical to the PCR-amplified one. The genome contains 3739 coding sequences, 41 tRNAs, and 4 rRNA genes (one each of 16S and 23S and two genes of 5S). A comparison between genome features of 10Alg 79^T^ and the type strains of closely related genera is presented in Table 1. These characteristics satisfy the proposed minimal standards for bacterial taxonomy [54].

To clarify genus-related features, a pan-genome analysis of 10Alg 79^T^ and representatives of three closely related genera (*M. arenosa* KCTC 52189^T^, *A. porphyridii* L1 8-17^T^, and *L. algarum* DSM 24677^T^) was performed using orthologous clustering with the OrthoVenn2 and OrthoVenn3 servers [41,42]. The analysis revealed 3374 orthologous gene clusters (gene families); 1602 single-copy clusters consisted of only single-copy orthologs presented in all lineages of the clade. The core and accessory (without unique genes) genomes consisted of 2167 and 1022 orthologous clusters, respectively (Figure 4). Most of the 10Alg 79^T^ clusters were shared with *M. arenosa* KCTC 52189^T^ (141) and *A. porphyridii* L1 8-17^T^ (116) (Figure 4). The results are in good agreement with the taxonomic position of 10Alg 79^T^ on the genomic tree (Figure 3); *M. arenosa* KCTC 52189^T^ and *A. porphyridii* L1 8-17^T^ were more related to 10Alg 79^T^ than *L. algarum* DSM 24677^T^. A total of 185 were identified as genus-specific clusters (Figure 4).

To predict metabolism of representatives of the nearest genera, genomes were first screened for the presence of genes of central and peripheral carbon metabolism due to the ability to grow on various carbon resources (Table 2). Based on the KEGG annotation, the genomes possess almost all genes of the glycolysis/gluconeogenesis pathways except for a key gene encoding 6-phosphofructokinase (EC 2.7.1.11) in the Embden–Meyerhof pathway. However, a modified pathway might incorporate the pentose phosphate pathway (PPP) due to genes encoding transaldolase (EC 2.2.1.2) and transketolase (EC 2.2.1.1). Moreover, genes found for the Entner–Doudoroff pathway (EDP), excluding *L. algarum* DSM 24677^T^, might encode the metabolism of glucose-6-phosphate into glyceraldehyde-3-phosphate and pyruvate. All the genomes were devoid of *pgd* encoding 6-phosphogluconate dehydrogenase (EC 1.1.1.44) in the oxidative phase of PPP. Semi-phosphorylative and non-phosphorilative EDP were presented by a partial set of genes. This means that the species are not able to utilize glucose through these canonical pathways and produce NADPH using PPP. All the genomes encode a complete tricarboxylic acid (TCA) cycle. Interestingly, *M. arenosa* KCTC 52189^T^ also encodes a complete Calvin–Benson–Bassham cycle, in which a key enzyme RuBisCO (EC 4.1.1.39), is encoded by *rbcLS* and accompanied by molecular chaperon genes *cbbQ* and *ccbO* to enhance CO_2_ fixation activity [55].

All four genomes shared genes and operons involved in phosphate metabolism, such as *pstSCAB* (high-affinity phosphate-specific transporter, TC 3.A.1.7.1), *phnCDE2E1* (phosphonate ABC transporter, TC 3.A.1.9.1), *phnGHIJKLNM* (carbon–phosphorus lyase enzyme complex), *ugpBAEC* (glycerol-3-phosphate transporter), *ugpQ* (cytosolic glycerophosphoryl diester phosphodiesterase, EC 3.1.4.46), *ppx* (exopolyphosphatase, EC 3.6.1.11), *ppa* (pyrophosphatase, EC 3.6.1.1), and *ppk* (polyphosphate kinase, EC 2.7.4.1).

All four genomes possessed a set of orthologous genes involved in ammonia assimilation, such as *gdhA* (NAD-specific glutamate dehydrogenase, 1.4.1.2; NADP-glutamate dehydrogenase, 1.4.1.4), *glnA* (glutamine synthetase, EC 6.3.1.2), *gltBD* (glutamate synthase, EC 1.4.1.13), *gltB* (ferredoxin-dependent glutamate synthase, EC 1.4.7.1), and *nadE* (ammonium-dependent NAD synthetase, EC 6.3.1.5). However, no genomes possessed the complete set of genes for denitrification; both the *A. porphyridii* L1 8-17^T^ and *M. arenosa* KCTC 52189^T^ genomes encode enzymes for nitrite, nitric and nitrous oxide reduction, while the 10Alg 79^T^ genome encodes only for nitrite, and nitric oxide reduction, whereas the *L. algarum* DSM 24677^T^ genome had none. Except for *L. algarum* DSM 24677^T^, most had *ncd2* (nitronate monooxygenase, EC 1.13.12.16), which is responsible for the oxidation of nitroalkane into nitrite.

In addition, to compare the carbon utilization potential, genes encoding CAZymes in 10Alg 79^T^, *A. porphyridii* L1 8-17^T^, *M. arenosa* KCTC 52189^T^, and *L. algarum* DSM 24677^T^ were predicted using the dbCAN2 server [43] (Figure 5). The 10Alg 79^T^ genome code 60 CAZymes, including 15 glycoside hydrolases (GHs), 36 glycosyltransferases (GTs), three carbohydrate esterases (CEs), seven auxiliary activities (AAs), and one carbohydrate-binding module (CBM).

Genome mining of 10Alg 79^T^ for secondary metabolite biosynthetic gene clusters (BGS) revealed five biotechnologically significant BGSs for the putative synthesis of non-ribosomal peptide synthetase (NRPS)-like and polyketide synthetase (PKS/NRPS hybrid), ribosomally synthesized and post-translationally modified peptides (RiPP-like, with methanobactin-like DUF692 domain), two homoserine lactones (hserlactone), and one important osmoprotectant ectoine. The most of these BGSs were also characteristic for the closely related genera *Marimonas*, *Lentibacter*, and *Aquicoccus*. For example, 10Alg 79^T^ sheared up to 90% of genes encoding the PKS/NRPS hybrid system with 50–88% of gene identities (Figure 6), as estimated by [44].

Out of 16,163 genes identified in the clade pan-genome, 2312 genes were assigned as singletons without any orthologues among the compared genomes. Among them 413 singletons were found in the 10Alg 79^T^ genome, followed by *L. algarum* DSM 24677^T^ (410), *A. porphyridii* L1 8-17^T^ (713), and *M. arenosa* KCTC 52189^T^ (775). An analysis of a unique part of each genome showed that most were annotated as hypothetical proteins (more than 60%). The annotated singletons were more often related to transport protein systems, transcriptional regulators, and enzymes.

To illustrate the novelty of the 10Alg 79^T^ genome, a detailed analysis of its singleton genome was carried out. In contrast to *A. porphyridii* L1 8-17^T^, *M. arenosa* KCTC 52189^T^, and *L. algarum* DSM 24677^T^, the 10Alg 79^T^ genome demonstrates the presence of additional genes encoding periplasmic glycerophosphodiester phosphodiesterase (GlpQ, EC 3.1.4.46), and sn-glycerol-3-phosphate (g3p) transport system permease (UgpA, TC 3.A.1.1.3) participating in transport and hydrolysis of organic phosphorus [56]. In addition, this genome contained six singleton genes and one orthologous gene encoding alkaline phosphatases (EC 3.1.3.1) indicating the role of mineralization in the transported organic phosphorus. For comparison, the *M. arenosa* KCTC 52189^T^ genome represents five orthologous genes and one singleton gene for alkaline phosphatases, followed by *L. algarum* DSM 24677^T^ with only two orthologous genes and *A. porphyridii* L1 8-17^T^ with only one orthologous gene. Therefore, attention should be paid to nitrogen metabolisms due to the number of singleton genes belonging to these subsystems. Remarkably, the 10Alg 79^T^ genome contains two urease operons for urea catabolism; the first *ureDABCEFG* was shared with those of the other three genomes, and the second *ureEFACGD* was composed of singleton genes homologous to one of the numerous urease operons of *Salipiger* spp. (about 80% amino acid identity). This second *ure* operon was accompanied by an additional urea transport operon *utrABCDE*, which is like that in *Salipiger* spp, indicating past lateral gene transfer events. Additionally, 10Alg79 harbors singletons encoding N-methylhydantoinase A and B (EC 3.5.2.14), involved in the metabolism of nitrogen-containing compounds.

Importantly, 10Alg 79^T^, in addition to *dddP*, has additional DMSP cleavage gene *dddL*, indicating its role in ocean DMSP metabolism. In addition, it can be speculated that singletons encoding CAZymes GH13_18 (EC 2.4.1.7), GH18 (EC 3.2.1.-), and GT81 (EC 2.4.1.266) provide genus-related features of 10Alg 79^T^. It should be noted that GH13_18 can be a 2-O-glucosylglycerate phosphorylase (GGaP; EC 2.4.1.352) based on the characteristic motifs of GGaP in a loop A (PYELN) and an acid/base loop (TETN) [57]. This novel phosphorylase catalyzes the reversible phosphorolysis of glucosylglycerate and has great potential for the synthesis of valuable phosphoglycosides [56].

### 3.3. Morphological, Physiological, and Biochemical Characteristics

Strain 10Alg 79^T^ has many properties in common with its closest relatives. They were found to be strictly aerobic, heterotrophic, and non-motile rods that can produce alkaline phosphatase, esterase (C4), esterase lipase (C8), leucine arylamidase, acid phosphatase, naphthol-AS-BI-phosphohydrolase, and catalase and oxidase but not lipase (C14), cystine arylamidase, trypsin, α-chymotrypsin, α-galactosidase, β-galactosidase, β-glucuronidase, α-glucosidase, β-glucosidase, N-acetyl-β-glucosaminidase, α-mannosidase, α-fucosidase, agarase, or caseinase. The novel isolate was distinguished from all neighbors by the minimal temperature for growth and amylase production. The sets of phenotypic traits including the ability to grow without NaCl or seawater, to reduce nitrate to nitrite, to produce acetoin and hydrogen sulfide, to hydrolyze aesculun, gelatin, tyrosine, and Tweens 40 and 80, taken together with the ability to utilize various carbohydrates, can help to discriminate strain 10Alg 79^T^ from each of its relatives (Table 2).

### 3.4. Chemotaxonomy

The fatty acid composition of strain 10Alg 79^T^ and reference strains *A. porphyridii* JCM 31543^T^ and *M. arenosa* KCTC 52189^T^ are listed in Table 3. The predominant fatty acids of strain 10Alg 79^T^ were C18:1 ω7c (82.1%) and C16:0 (8.2%) (Table 3). The fatty acid composition of the strain studied was similar to that of the reference strains, although differences were found in the proportions of some fatty acids (Table 3). The polar lipid profile of strain 10Alg 79^T^ contained phosphatidylethanolamine, phosphatidylglycerol, phosphatidylcholine, an unidentified phospholipid, an unidentified aminolipid, and an unidentified lipid and was consistent with that of the reference species (Appendix A; Table 3). However, the presence of an unidentified phospholipid distinguished the bacterial isolate from *M. arenosa* KCTC 52189^T^ and *L. algarum* ZXM100^T^. Moreover, strain 10Alg 79^T^ can be differentiated from *M. arenosa* KCTC 52189^T^ by the absence of an unknown glycolipid (Table 3). The predominant respiratory quinone was Q-10.

## 4. Conclusions


**Description of *Rhodoalgimonas* gen. nov.**


*Rhodoalgimonas* (Rho.do.al.gi.mo’nas. Gr. adj. rhodos red (rose); L. fem. n. alga seaweed, L. fem. n. monas a monad, a unit; N.L. fem. n. *Rhodoalgimonas*, a red (rose) monad isolated from seaweed).

Cells are Gram-stain-negative, strictly aerobic, rod-shaped, non-spore-forming, and non-motile. Catalase- and oxidase-positive. The dominant fatty acids (>5%) are summed feature 8, comprising C18:1 ω7c and C16:0. The polar lipids are phosphatidylethanolamine, phosphatidylglycerol, phosphatidylcholine, unidentified aminolipid, unidentified phospholipid and unidentified lipid. The major respiratory quinone is Q-10. Phylogenetically, the genus belongs to the family *Rhodobacteraceae*, the class *Alphaproteobacteria*, and the phylum *Pseudomonadota*. The type species is *Rhodoalgimonas zhirmunskyi*


**Description of *Rhodoalgimonas zhirmunskyi* sp. nov.**


*Rhodoalgimonas zhirmunskyi* (zhir.mun’skyi. N.L. gen. masc. n. *zhirmunskyi* named in honor of the academician A.V. Zhirmunskyi for his great contribution to the development of marine biology in the Russian Far East).

Cells are heterotrophic, strictly aerobic, non-motile, Gram-stain-negative rods, and 0.3–0.7 μm wide and 0.8–2.1 μm long. On marine agar, colonies are circular, shiny, slightly convex, with entire edges, 1–2 mm in diameter, and beige-colored. Growth occurs at 4–40 °C (optimum is 30–32 °C) and pH 6.0–9.0 (optimum is 7.0–8.0) and with 0–5% NaCl (optimum is 1.5–3% NaCl). Seawater or artificial seawater is substantial for growth. Catalase and oxidase activities are present. Arginine dihydrolase, lysine decarboxylase, ornithine decarboxylase, and tryptophan deaminase activities are absent. Aesculin, starch, and Tweens 20, 40, and 80 are hydrolyzed but agar, casein, gelatin, chitin, CM-cellulose, DNA, L-tyrosine, and urea are not. Acid is not produced from L-arabinose, D-cellobiose, D-fructose, D-galactose, D-glucose, D-lactose, maltose, mannose, melibiose, raffinose, L-rhamnose, ribose, sucrose, trehalose, D-xylose, N-acetylglucosamine, mannitol, glycerol, sorbitol, or citrate. Growth is observed on L-arabinose, D-cellebiose, D-fructose, D-galactose, D-glucose, D-lactose, maltose, D-mannose, D-melibiose, D-raffinose, L-rhamnose, D-trehalose, D-xylose, N-acetylglucosamine, L-alanine, L-asparagine, L-histidine, L-methionine, L-proline, L-triptophane, L-treonine, L-tyrosin, and L-valine. In the API 20NE gallery, positive for aesculin hydrolysis and assimilation of D-mannose, D-mannitol, maltose, gluconate, adipate, malate, and phenylacetate tests. In the API 20E and API 50 CH kits, no results were positive. In the API ID 32GN kit, maltose, D-mannitol, salicin, D-melibiose, propionic acid, valeric acid, and 3-hydroxybutiric acid were assimilated. According to the API ZYM tests, alkaline phosphatase, esterase (C4), esterase lipase (C8), leucine arylamidase, acid phosphatase, and naphtol-AS-BI-phosphohydrolase activities are present but lipase (C14), cystine arylamidase, trypsin, *α*-chymotrypsin, *α*-galactosidase, *β*-galactosidase, *β*-glucuronidase, *α*-glucosidase, *β*-glucosidase, N-acetyl-*β*-glucosaminidase, *α*-mannosidase, and *α*-fucosidase activities are absent. Valine arylamidase activity was weakly positive. Nitrate is not reduced. Hydrogen sulfide, indole, and acetoin are not produced. The prevalent fatty acids are summed feature 8, comprising C18:1 ω7c and C16:0. The polar lipid profile consists of phosphatidylethanolamine, phosphatidylglycerol, phosphatidylcholine, an unidentified aminolipid, an unidentified phospholipid, and an unidentified lipid. The main ubiquinone is Q-10. The genomic DNA GC content of the type strain is 62.1 mol%.

## Figures and Tables

**Figure 1 microorganisms-11-02463-f001:**
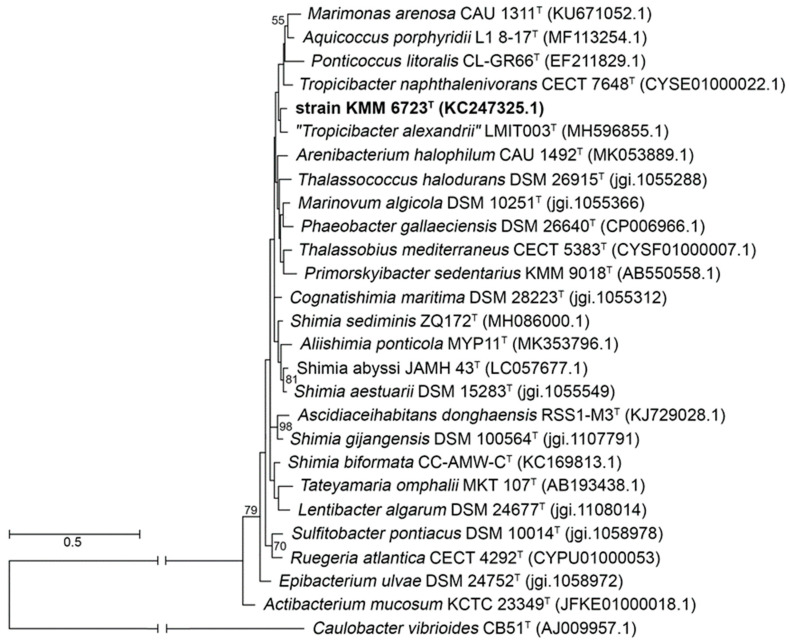
ML 16S rRNA tree showing phylogenetic relationships between the novel strain 10Alg 79^T^ (KMM 6723^T^) and members of the family *Rosebacteraceae* (formerly *Rhodobacteriaceae*). Bootstrap values shown as percentage are based on 500 replicates. Bars are 0.5 substitutions per nucleotide. Strain *Caulobacter vibrioides* CB51^T^/DSM 9893^T^ was used as an outgroup. GenBank/EMBL/DDB accession numbers are given in parentheses.

**Figure 2 microorganisms-11-02463-f002:**
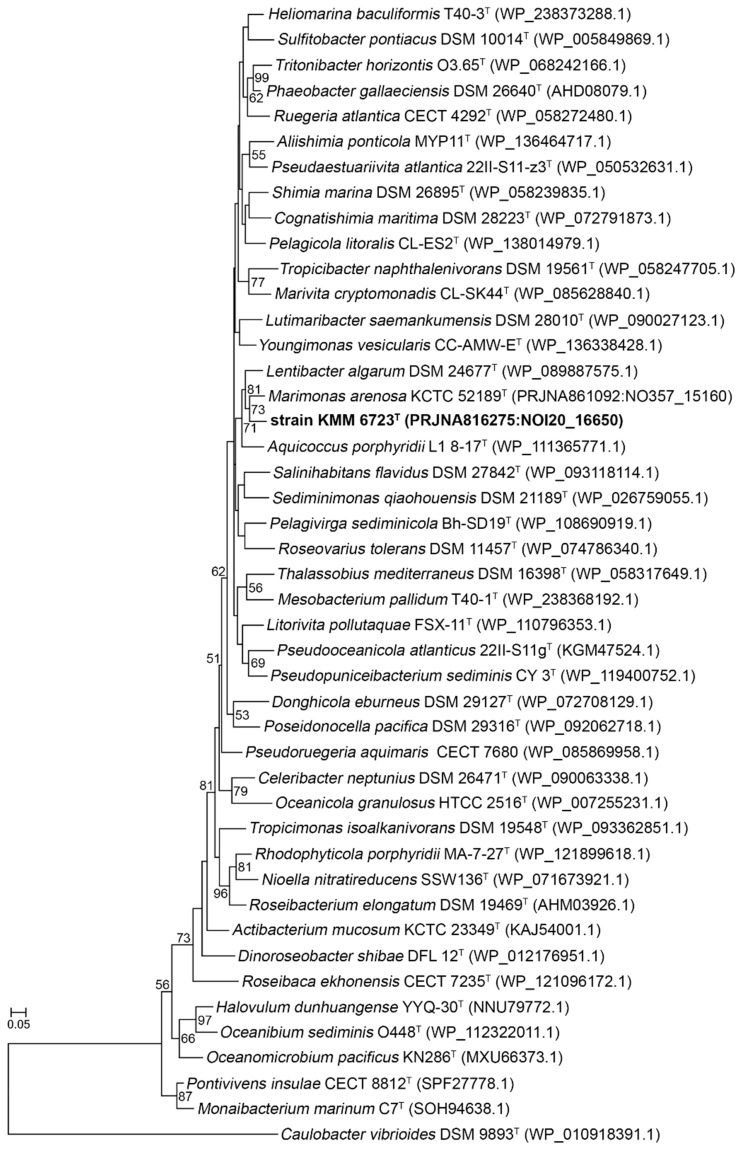
ML RpoC tree showing phylogenetic relationships between the novel strain 10Alg 79^T^ (KMM 6723^T^) and 43 members of the family *Rosebacteraceae* (formerly *Rhodobacteriaceae*). Bootstrap values are based on 500 replicates and shown as percentages greater than 50. Bars are 0.5 substitutions per amino acid position. Strain *Caulobacter vibrioides* DSM 9893^T^ was used as an outgroup. GeBank/EMBL/DDB accession numbers are given in parentheses.

**Figure 3 microorganisms-11-02463-f003:**
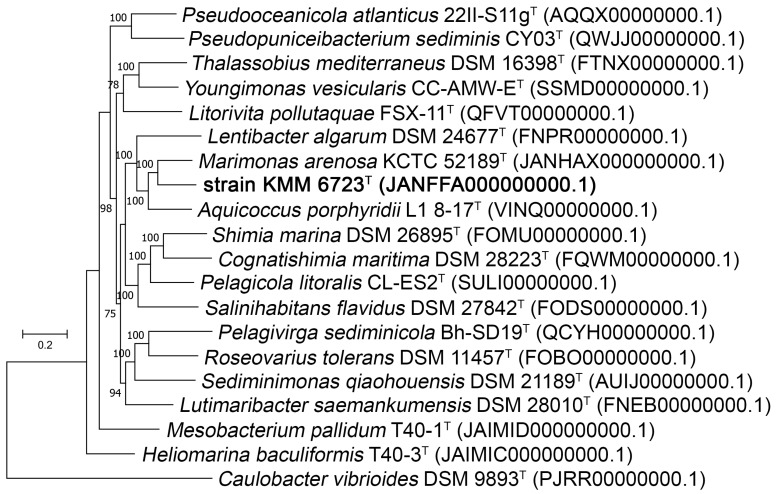
ML genomic tree showing phylogenetic relationships between the strain 10Alg 79^T^ (KMM 6723^T^) and members of the family *Rosebacteraceae* (formerly *Rhodobacteriaceae*). Bootstrap values shown as percentages greater than 50 are based on 100 replicates. Bar is 0.2 substitutions per amino acid position. Strain *Caulobacter vibrioides* DSM 9893^T^ was used as an outgroup. GenBank/EMBL/DDB accession numbers are given in parentheses.

**Figure 4 microorganisms-11-02463-f004:**
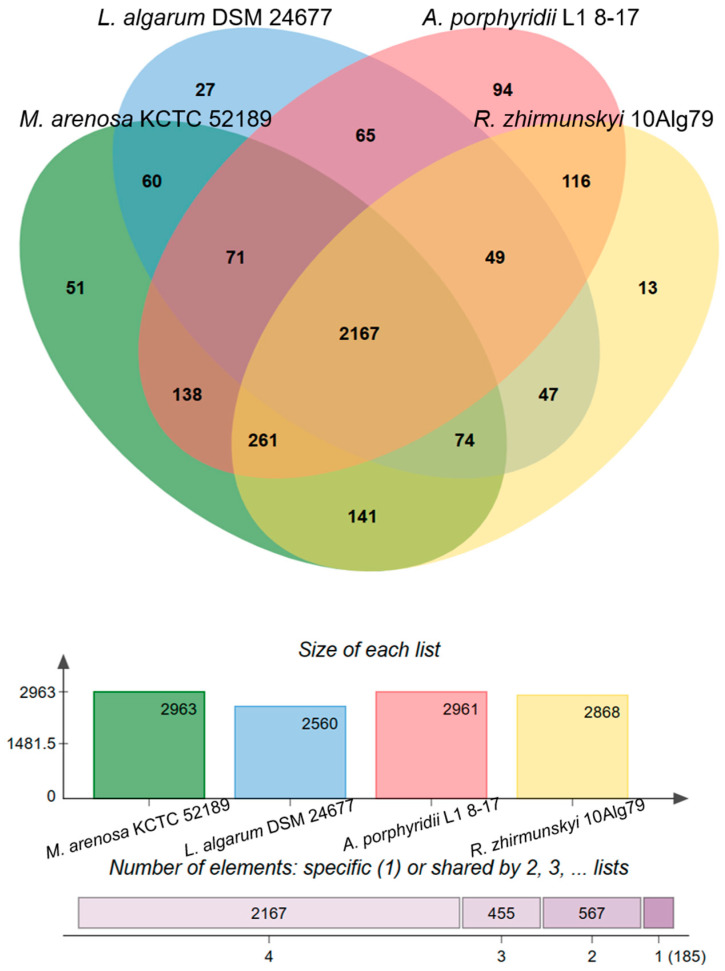
Venn diagram showing the distribution of shared orthologous clusters among genomes of 10Alg 79^T^, *A. porphyridii* L1 8-17^T^, *M. arenosa* KCTC 52189^T^, and *L. algarum* DSM 24677^T^. The orthologous clusters are comprised of proteins predicted using RAST [36]. The number of orthologous clusters for each genome is shown in vertical bar chart.

**Figure 5 microorganisms-11-02463-f005:**
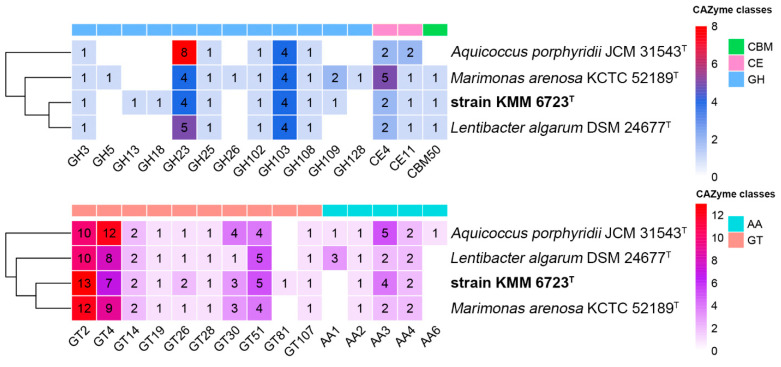
The distribution of CAZymes across genomes of KMM 6723^T^ (=10Alg 79^T^) and close relatives. The heat maps show the number of genes assigned to individual CAZyme families. Rows are clustered using Euclidean distances. GH, glycoside hydrolases; GT, glycosyltransferases, CE, carbohydrate esterase; AA, auxiliary activity; CBM, carbohydrate-binding module.

**Figure 6 microorganisms-11-02463-f006:**
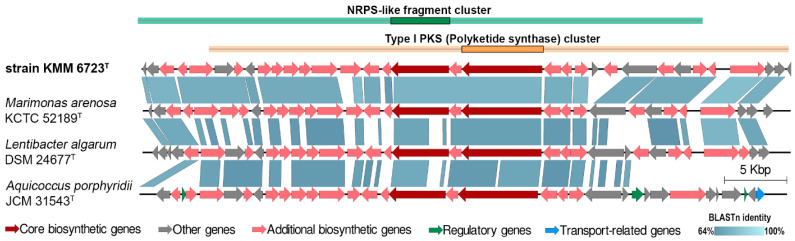
Comparative analysis of synteny between NRPS-like fragment and Type I PKS BGCs in the genomes of strain 10Alg 79^T^ (KMM 6723^T^) and close relatives. Genes are colored based on their annotation as indicated at the bottom. The locus regions are NOI20_03420 to NOI20_03580 in the KMM 6723^T^ genome, NO357_05410 to NO357_05560 in the *M. arenosa* KCTC 52189^T^ genome, BLV93_RS04080 to BLV93_RS03920 in the *L. algarum* DSM 24677^T^ genome, and FLO80_RS04395 to FLO80_RS04560 in the *A. porphyridii* JCM 31543^T^ genome.

**Table 1 microorganisms-11-02463-t001:** Genomic features of 10Alg 79^T^ and three type strains of closely related genera.

Feature	1	2	3	4
Assembly level	Contig	Contig	Contig	Contig
Genome size (bp)	3,754,741	4,513,837	4,380,839	3,293,487
Number of contigs	73	256	38	21
GC content (mol%)	62.1	63.2	63	55.5
N50 (bp)	702,161	179,538	656,426	890,063
L50 (bp)	2	7	3	2
Coverage	19x	100x	90x	311.0x
Total genes	3609	4542	4299	3318
Protein coding genes	3535	4373	4205	3262
rRNAs(5S/16S/23S)	2/1/1	2/1/1	1/1/1	62/2/2
tRNA	41	42	44	44
checkM completeness (%)	98.24	100	99.42	99.69
checkM contamination (%)	0.04	2.6	0.34	0.31
WGS project	JANFFA01	VINQ01	JANHAX01	FNPR01
Annotated genome assembly	ASM3084892v1	ASM836910v1	ASM3084894v1	IMG-taxon 2693429861
Submitted GenBank assembly	GCA_030848925.1	GCA_008369105.1	GCA_030848945.1	GCA_900107355.1
Number of Subsystems (RAST)	300	324	315	285

Strains: **1**, 10Alg 79^T^; **2**, *Aquicoccus porphyridii* L1 8-17^T^; **3**, *Marimonas arenosa* KCTC 52189^T^; **4**, *Lentibacter algarum* DSM 24677^T^.

**Table 2 microorganisms-11-02463-t002:** Phenotypic characteristics differentiating strain 10Alg 79^T^ and related genera of the family *Roseobacteraceae*.

Characteristics	1	2	3	4
Source of isolation	Marine red alga	Marine red alga	Sea sand	Seawater
Colony color	Beige	Whitish	Beige	Whitish
Morphology	Rod	Coccus	Rod	Rod
Temperature range for growth (°C) *:	4–40	20–40	20–37	22–28
Salinity range for growth (% NaCl) *:	0–5	0–7	0–6	3–9
Nitrate reduction	-	-	-	+
Acetoin production (3-hydroxybutanone or acetyl methyl carbinol)	-	-	-	+
H_2_S production	-	+	-	-
Degradation of:				
Aesculin	W	-	-	+
Gelatin	-	-	+	-
Starch	W	-	-	-
Tweens 40 and 80	+	-	-	+
Tyrosine	-	+	-	ND
Acid production from:				
D-fructose, D-glucose, maltose, sucrose, D-xylose	-	-	-	+
Utilization of:				
L-arabinose, maltose	+	+	-	+
D-cellobiose, D-fructose,	+	-	-	+
D-lactose, D-melibiose	+	-	-	ND
D-glucose, D-mannose	+	-	-	+
D-mannitol	+	-	+	+
N-acetylglucosamine	+	-	-	-
Adipate	+	-	+	-
Citrate	-	-	-	+
Gluconate	+	-	+	+
Malate	+	+	+	-
Phenylacetate	+	+	-	-
Susceptibility to:				
Ampicillin, carbenicillin	+	-	+	+
Gentamicin	+	-	+	-
Bensylpenicillin	+	-	-	+
Kanamycin	+	+	+	-
Oxacillin	+	-	-	-
Tetracycline	+	+	-	-
Vancomycin	+	+	-	+
Major polar lipids **	PC, PG, PE, AL, PL, L	PC, PG, PE, AL, PL, L	PC, PG, PE, AL, GL	PC, PG, PE, AL, L

Strains: **1**, 10Alg 79^T^ (this study); **2**, *Aquicoccus porphyridii* JCM 31543^T^ (this study); **3**, *Marimonas arenosa* KCTC 52189^T^ (this study); **4**, *Lentibacter algarum* ZXM100^T^ (data from [50]). All strains were positive for the following tests: respiratory type of metabolism; alkaline phosphatase, esterase (C4), esterase lipase (C8), leucine arylamidase, acid phosphatase, naphthol-AS-BI-phosphohydrolase, catalase and oxidase activities; susceptibility to erythromycin, and resistance to polymixin. All strains were negative for the following tests: motility; indole production; hydrolysis of agar, casein; acid production from D-lactose, D-melibiose, L-rhamnose, and ribose; lipase (C14), cystine arylamidase, trypsin, *α*-chymotrypsin, α-galactosidase, *β*-galactosidase, *β*-glucuronidase, *α*-glucosidase, *β*-glucosidase, N-acetyl-*β*-glucosaminidase, *α*-mannosidase and *α*-fucosidase activities. +, Positive; -, negative; W, a weak reaction; ND, data are not detected. *, Data taken from [48,49] **, PC, phosphatidylcholine; PG, phosphatidylglycerol; PE, phosphatidylethanolamine; AL, unidentified aminolipid; PL, unidentified phospholipid; L, unidentified lipid; and GL, unidentified glycolipid.

**Table 3 microorganisms-11-02463-t003:** Fatty acid profiles of strain 10Alg 79^T^ and the type strains of the related genera of the family *Roseobacteraceae*.

Fatty Acid	1	2	3	4
Straight fatty acid				
C_15:0_	1.3	tr	-	-
C_16:0_	8.2	7.7	4.5	12.0
C_17:0_	1.6	1.5	-	2.0
C_18:0_	Tr	2.3	3.9	20.0
Unsaturated fatty acid				
C_16:1_ *ω7c*	1.3	tr	Tr	1.0
C_18:1_ *ω7c*	82.1	81.8	80.6	60.0
11-methyl C_18:1_ *ω7c*	Tr	4.3	4.1	2.0
Hydroxy fatty acid				
C_10:0_ 3-OH	-	-	-	2.0
C_12:0_ 3-OH	4.5	1.0	4.0	-
C_12:1_ 3-OH	-	1.2	-	-

Strains: **1**, 10Alg 79^T^; **2**, *Aquicoccus porphyridii* JCM 31543^T^; **3**, *Marimonas arenosa* KCTC 52189^T^; **4**, *Lentibacter algarum* ZXM100^T^ (data from [48]). Data are from the present study. Values are percentages of the total fatty acids. tr, trace amount (<1%).

## Data Availability

The type strain of the species is strain 10Alg 79^T^ isolated from the red alga *Ahnfeltia tobuchiensis* collected near Island Paramushir, Kuril Isles, the Sea of Okhotsk, Pacific Ocean, Russia. The DDBJ/ENA/GenBank accession numbers for the 16S rRNA gene and the whole-genome shotgun sequences are KC247325 and JANFFA000000000, respectively. Strain 10Alg 79^T^ was deposited in the Collection of Marine Microorganisms (WFCC acronym is KMM) under the number KMM 6723^T^, and in the Korean Collection for Type Cultures (KCTC) under the number KACC 72611^T^.

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
