# Peer review of "Rhodoalgimonas zhirmunskyi gen. nov., sp. nov., a Marine Alphaproteobacterium Isolated from the Pacific Red Alga Ahnfeltia tobuchiensis: Phenotypic Characterization and Pan-Genome Analysis"

_microorganisms, 2023, doi:10.3390/microorganisms11102463_

Round 1
Reviewer 1 Report
In this manuscript, authors proposed Rhodoalgimonas zhirmunskyi gen. nov., sp. nov., based on 16S rRNA gene sequence, genomic analysis of ANI, AAI, dDDH and G+C content, and chemotaxonomic analysis such as fatty acids and polar lipid profile, and physiological characteristics. Authors also showed pan-genome analysis of related genera, and identified some singletons in strain 10Alg 79.
The manuscript is well written, and the data is enough to support the proposal of new genus.
Comments:
1) The main focus of this manuscript is proposal of novel genus. It is not matched with the title. Please modify the title.
2) Authors need to add some experiments to confirm the genomic characteristics of strain 10Alg 79. For example, authors said “10Alg 79T genome contains two urease operons for urea catabolism” (Lines 336-337), and I think authors easily check urease activity with experiments.
Minor comments:
1) Line 24: please change “97,33%” to “97.33%”
2) Line 143: “Illumina MiSeq platform with a 150-bp read length”. Is it correct? Normally, we use Illumina Hiseq platform for 150-bp read length, but Illumina Miseq platform for 250-bp or 300-bp read length.
3) Line 397: the “10Alg 79T” should be written as “10Alg 79T”.
Author Response
Responses for Reviewer 1
Comment: In this manuscript, authors proposed Rhodoalgimonas zhirmunskyi gen. nov., sp. nov., based on 16S rRNA gene sequence, genomic analysis of ANI, AAI, dDDH and G+C content, and chemotaxonomic analysis such as fatty acids and polar lipid profile, and physiological characteristics. Authors also showed pan-genome analysis of related genera, and identified some singletons in strain 10Alg 79. The manuscript is well written, and the data is enough to support the proposal of new genus.
Response. Thank you very much for reviewing our manuscript. We appreciate your valuable comments and suggestions. We have revised our manuscript in accordance with your comments.
Comment: The main focus of this manuscript is proposal of novel genus. It is not matched with the title. Please modify the title.
Response. Thank you for еру suggestion. We have changed the title of our manuscript as «Rhodoalgimonas zhirmunskyi gen. nov., sp. nov., a marine alphaproteobacterium isolated from the Pacific red alga Ahnfeltia tobuchiensis: phenotypic characterization and pan-genome analysis».
Comment: Authors need to add some experiments to confirm the genomic characteristics of strain 10Alg 79. For example, authors said “10Alg 79T genome contains two urease operons for urea catabolism” (Lines 336-337), and I think authors easily check urease activity with experiments.
Response. Urease activity was not observed for 10Alg 79 (Line 517) as well as it was reported for other compared relatives Marimonas arenosa KCTC 52189T and Marimonas lutisalis GH1-19T. The activity was reported for the type strain Lentibacter algarum LMG 24861T. Aquicoccus porphyridii L1 8-17T has not be tested. All the genomes encode at least one urease operon. It can be assumed that the urease operons might be turned on by unknown stimuli. We plan to study this phenomenon in bacteria associated with macroalgae, which can play an important role in marine nitrogen cycling.
Li, Z.; Qu, Z. Lentibacter algarum gen. nov., sp. nov., isolated from coastal water during a massive green algae bloom. Int. J. Syst. Evol. Microbiol. 2012, 62, 1042-1047.
Thongphrom, C.; Kim, J.H. Marimonas arenosa gen. nov., sp. nov., isolated from sea sand. Int. J. Syst. Evol. Microbiol. 2017, 67, 121-126.
Lee SD, Jeon D, Kim YJ, Kim IS, Choe H, Kim JS. Marimonas lutisalis sp. nov., isolated from a tidal mudflat and emended description of the genus Marimonas. Int J Syst Evol Microbiol. 2020 Jan;70(1):259-266. doi: 10.1099/ijsem.0.003749. PMID: 31639073.
Feng, T.; Kim, K.H. Aquicoccus porphyridii gen. nov., sp. nov., isolated from a small marine red alga, Porphyridium marinum. Int. J. Syst. Evol. Microbiol. 2018, 68, 283-288.
Minor comments:
1) Line 24: please change “97,33%” to “97.33%”
Response: Thank you very much. It was corrected (Line 26).
2) Line 143: “Illumina MiSeq platform with a 150-bp read length”. Is it correct? Normally, we use Illumina Hiseq platform for 150-bp read length, but Illumina Miseq platform for 250-bp or 300-bp read length.
Response: Please, look at MiSeq Reagent Kit v2, Read Length is 2 × 150 bp (https://www.illumina.com/systems/sequencing-platforms/miseq/specifications.html)
3) Line 397: the “10Alg 79T” should be written as “10Alg 79T”.
Response: Thank you very much. It has been corrected and checked throughout the manuscript.
Reviewer 2 Report
The study conducted by Nedashkovskaya et al. reported the genomic characterization and taxonomic description of novel isolated bacterium “10Alg 79T” from red alga, Ahnfeltia tobuchiensis. Actually, this is an interesting, good work but need minor corrections.
The title is strong as a title, however it not represent the work or in another words the MS especially abstract not reflect the title, such as the ecological and biotechnological potentiality, could you express the relation between the bacterium characterization and potentiality.
Line 36: correct “algal isolate” to “bacterium isolated”, considered along the MS.
Could you give more information about the location in the M and M “Sea of Okhotsk” in which country (Line 97).
Table 3: could you present the data as mean ± Standard error.
The conclusion is very long, it was repetition of the full results, good conclusion must be a take home message with general trend of the results. For example, we did not need to mention all isolated amino acids but we can mention if it has a high level of essential amino acids.
Minor editing of English language required.
Author Response
Responses for Reviewer 2
Comment: The study conducted by Nedashkovskaya et al. reported the genomic characterization and taxonomic description of novel isolated bacterium “10Alg 79T” from red alga, Ahnfeltia tobuchiensis. Actually, this is an interesting, good work but need minor corrections.
Response. Thank you very much for reviewing our manuscript. We appreciate your valuable comments and suggestions. We have revised our manuscript in accordance with your comments.
Comment: The title is strong as a title, however it not represent the work or in another words the MS especially abstract not reflect the title, such as the ecological and biotechnological potentiality, could you express the relation between the bacterium characterization and potentiality.
Response: Thank you for your comment. We have changed the title of our manuscript and added several sentences about the relationship between the characteristics of the bacterium and its potential (Lines 36-39).
Comment: Line 36: correct “algal isolate” to “bacterium isolated”, considered along the MS.
Response: Thank you very much. It was corrected (Lines 39, 468).
Comment: Could you give more information about the location in the M and M “Sea of Okhotsk” in which country (Line 97).
Response: Thank you very much. It was added (Line 106).
Comment: Table 3: could you present the data as mean ± Standard error.
Response: As usual, these data are presented without “± Standard error”. We adhere to the standards required for the description of new taxa.
Comment: The conclusion is very long, it was repetition of the full results, good conclusion must be a take home message with general trend of the results. For example, we did not need to mention all isolated amino acids but we can mention if it has a high level of essential amino acids.
Response: The Conclusion section presents the mandatory part of description and named of new taxon. “The aim is to standardize the format of descriptions of new taxa. The properties of the taxon being described must be given directly after (a) the new name or new combination (i.e. fam. nov., sp. nov. etc.) and (b) the derivation of a new name” (Reference: DE VOS (P.) and TRÜPER (H.G.): Judicial Commission of the International Committee on Systematic Bacteriology. IXth International (IUMS) Congress of Bacteriology and Applied Microbiology. Minutes of the meetings, 14, 15 and 18 August 1999, Sydney, Australia. Int. J. Syst. Evol. Microbiol. 2000, 50, 2239-2244.).
Reviewer 3 Report
Review on “Genomic characterization of Rhodoalgimonas zhirmunskyi gen. nov., sp. nov., a novel rare marine alphaproteobacterium, reveals biosynthetic, ecological, and biotechnological potential” for Microorganisms (manuscript ID microorganisms-2590737)
In this manuscript the authors describe the novel bacterial strain associates the red alga Ahnfeltia tobuchirnsis. Despite the actual topic the manuscript requires multiple improvements.
The most of Introduction section devoted to plain description of known bacterial species associated with the red algae. It’s better to describe the importance of particular isolate (10Alg79) study, what the purpose of the present work behind the routine genome report.
L25: in abstract nearest neighbor defined as Shimia sediminis ZQ172, but further in the main text we see different strains (L184). Details of the phylogenetic relationships are redundant in the abstract.
L30. L449: what “mol%” stands for?
My questions about Results and Discussion:
L188: most of the bootstrap values are missing in the tree (Figure 1A), so it’s hard to evaluate the tree consistency. It might help to include high resolution tree and source files in the Supplementary.
L204: what criteria was used to select 18 species for tree building with PhyloPhlAn ? If rRNA-based tree is inconsistent (poor bootstrap scores), why we considering it at all? The whole-genome approach initially can be used if genomes are available.
L200: As it known, PhyloPhlAn don’t use the bootstrap by default, because whole-genome tree is already consistent (see Fig. 2 in https://www.nature.com/articles/s41467-020-16366-7)
L236: Genome statistics is different in the manuscript and the NCBI genome assembly page for the KCTC 52189 https://www.ncbi.nlm.nih.gov/assembly/GCF_030848945.1/
Figure 1. What highlighting at the trees stands for? Strains selection criteria is unclear.
Figure 2. What is the purpose of “Number of elements…” caption?
L368: Table 2. The part of the legend might be included in the table itself: “All strains were negative for the following tests: …”
L396: “glycolipid” is missing in the Table 1 and other tables too.
Part of experimental data in the Results and Discussion section are given “as is” without comparison and interpretation with similar studies.
The Conclusion section should contain the brief summary of the key findings of the study, please rewrite it from scratch.
Methods section comments:
L95: If cell morphology was studied using TEM, I suggest to add the figure with the cells image.
Why the particular strains were used for the comparison?
L117: “Aquicoccus porphyridii JCM 31543T and Marimonas arenosa KCTC 52189T, were extracted and analyzed as described previously [23]”, but this paper covers different strain (KMM 6059), not the 10Alg79/KMM6723.
L174: “Pairwise comparisons of each locus between ten genomes…”, but only 4 genomes are shown at Figure 4. What 10 genomes were used for comparison?
L179: is the genome of previously described strain (M. arenosa KCTC 52189T) is available? First description is dated 2017 https://doi.org/10.1099/ijsem.0.001581
The name of the strain 10Alg79 and collection ID KMM6723 both used in the manuscript, that could be misleading.
Some minor corrections to the text (style and spelling):
· L55, L185: the backticks could be removed

Author Response
Responses for Reviewer 3
Comment: Review on “Genomic characterization of Rhodoalgimonas zhirmunskyi gen. nov., sp. nov., a novel rare marine alphaproteobacterium, reveals biosynthetic, ecological, and biotechnological potential” for Microorganisms (manuscript ID microorganisms-2590737)
In this manuscript, the authors describe the novel bacterial strain associates the red alga Ahnfeltia tobuchirnsis. Despite the actual topic the manuscript requires multiple improvements.
The most of Introduction section devoted to plain description of known bacterial species associated with the red algae. It’s better to describe the importance of particular isolate (10Alg79) study, what the purpose of the present work behind the routine genome report.
Response: The introduction of this manuscript provides information on which new taxa have been isolated from red algae, including members of the Alphaproteobacteria. We believe that this analysis is an important aspect for understanding the diversity of algae-associated bacteria, and predicting their potential for the discovery of new enzymes.
Comment: L25: in abstract nearest neighbor defined as Shimia sediminis ZQ172, but further in the main text we see different strains (L184). Details of the phylogenetic relationships are redundant in the abstract.
Response: One point of the mandatory information when describing a new taxon is the indication of 16S rRNA pairwise similarity values. Shimia sediminis ZQ172 is the closest neighbor of the new strain 10Alg 79 based on 16S rRNA pairwise similarity values. However, ZQ172 is situated on a separate lineage which is significantly distant to the 10Alg 79 in the 16S rRNA tree. In other phylogenetic trees (Figures 2 and 3) Shimia sediminis was not used because a type species of the genus is Shimia marina.
Comment: L30. L449: what “mol%” stands for?
Response: The value of the G+C mol% content is based on the genomic sequence of the strain.
Comment: My questions about Results and Discussion: L188: most of the bootstrap values are missing in the tree (Figure 1A), so it’s hard to evaluate the tree consistency. It might help to include high resolution tree and source files in the Supplementary.
Response: The bootstrap values are not missing in the 16S rRNA tree. They were not shown because their confidence was low. We indicated this in the caption to the Figure 1 as “Bootstrap values shown as percentage are based on 500 replicates.” We have decided to present this tree as a separate Figure. Please, see Figure 1.
Comment: L204: what criteria was used to select 18 species for tree building with PhyloPhlAn ? If rRNA-based tree is inconsistent (poor bootstrap scores), why we considering it at all? The whole-genome approach initially can be used if genomes are available.
Comment: The rRNA-based tree does not help select relatives for phylogenomic analysis. To do this, we took another phylogenetic gene marker as rpoC. For clarity, we have moved Figure S1 with RpoC-based tree from the Supplementary to the Results section. Please, see Figure 2.
Comment: L200: As it known, PhyloPhlAn don’t use the bootstrap by default, because whole-genome tree is already consistent (see Fig. 2 in https://www.nature.com/articles/s41467-020-16366-7)
Response: Thank you for your remark. Indeed, as a part of PhyloPhlAn pipeline, RAxML program builds phylogenetic trees without bootstrapping by default. We just added such option in config file.
Comment: L236: Genome statistics is different in the manuscript and the NCBI genome assembly page for the KCTC 52189 https://www.ncbi.nlm.nih.gov/assembly/GCF_030848945.1/
Response: We agree with your comment. These have been corrected. Please, see Table 1.
Comment: Figure 1. What highlighting at the trees stands for? Strains selection criteria is unclear.
Response: The purpose was to highlight the strains that are closely related by genomes or similarity to the new strain but were distant on the 16S rRNA tree. To avoid confusion, we have removed the color highlighting of these strains. Please, see Figure 1 and 3.
Comment: Figure 2. What is the purpose of “Number of elements…” caption?
Response: The number of elements indicated: (1) the sum of taxon-specific orthologous clusters occurring in one genome; (2) and (3) the sum of orthologous clusters occurring in two or three genomes.
Comment: L368: Table 2. The part of the legend might be included in the table itself: “All strains were negative for the following tests: …”
Response: The table 2 states the differences between strains. Therefore, the identical characteristics were left out.
Comment: L396: “glycolipid” is missing in the Table 1 and other tables too.
Response: Thank you very much for the remark. This has been fixed. Table S1 has been added to the Supplementary.
Comment: Part of experimental data in the Results and Discussion section are given “as is” without comparison and interpretation with similar studies.
Response: This comment is not clear. Most of comparisons are presented in tabular form.
Comment: The Conclusion section should contain the brief summary of the key findings of the study, please rewrite it from scratch.
Response: The Conclusion section presents the mandatory part of description and named of new taxon. “The aim is to standardize the format of descriptions of new taxa. The properties of the taxon being described must be given directly after (a) the new name or new combination (i.e. fam. nov., sp. nov. etc.) and (b) the derivation of a new name” (Reference: DE VOS (P.) and TRÜPER (H.G.): Judicial Commission of the International Committee on Systematic Bacteriology. IXth International (IUMS) Congress of Bacteriology and Applied Microbiology. Minutes of the meetings, 14, 15 and 18 August 1999, Sydney, Australia. Int. J. Syst. Evol. Microbiol. 2000, 50, 2239-2244.).
Methods section comments:
Comment: L95: If cell morphology was studied using TEM, I suggest to add the figure with the cells image.
Response: Unfortunately, the cells image were lost.
Comment: Why the particular strains were used for the comparison?
Response: Since the new strain 10Alg 79 is potentially a new species of a new genus, the type strains of type species of closely related genera were taken for comparison.
Comment: L117: “Aquicoccus porphyridii JCM 31543T and Marimonas arenosa KCTC 52189T, were extracted and analyzed as described previously [23]”, but this paper covers different strain (KMM 6059), not the 10Alg79/KMM6723.
Response: This comment is not clear. In this manuscript, esters and lipids were prepared and analyzed in the same way as in ref. 23.
Comment: L174: “Pairwise comparisons of each locus between ten genomes…”, but only 4 genomes are shown at Figure 4. What 10 genomes were used for comparison?
Response: Thank you very much for the remark. This has been fixed (Line 185).
Comment: L179: is the genome of previously described strain (M. arenosa KCTC 52189T) is available? First description is dated 2017 https://doi.org/10.1099/ijsem.0.001581
Response: The draft genome of M. arenosa KCTC 52189T was obtained in this study as described above for the draft genome of 10Alg 79T and used in the comparative genome analysis (Lines 189-190).
Comment: The name of the strain 10Alg79 and collection ID KMM6723 both used in the manuscript that could be misleading.
Response: It is common practice to use a working (original) name and a collection name for strain together.
Comment: Some minor corrections to the text (style and spelling):
- L55, L185: the backticks could be removed
Response: Nomenclatural status for Psychroserpens luteolus Ping et al. 2023 is: validly published. The backticks were removed. Nomenclatural status for "Tropicibacter alexandrii" Wang et al. 2020 is: not validly published. The backticks are preserved.
Round 2
Reviewer 3 Report
I would to thank authors for the efforts to improve the manuscript.
Some minor issues are have to be resolved:
L205: how the sequence of RpoC gene was obtained? It was amplified with primers or derived from whole genome assembly?
In Table 1 I recommend to refer to NCBI assembly IDs, not names and add the hyperlinks: GCA_030848925.1 instead of ASM3084892v1 and so on, because ASM3084892v1 cannot be found in NCBI website.
L359: "clade`" apostrophe is redundant here
L396: "note" to "noted"
L414: "hydrolase" maybe "hydrolyse" instead?
L415: "Tweens 40 and 80" are not substance names but commercial reagents
Author Response
Replies.
- In Table 1 we have added hyperlinks on NCBI submitted assemblies.
- RpoC gene sequences were derived from whole genome assembly. Lines 203-205. "Therefore, the position of strain 10Alg 79T was further determined using rpoC gene sequences extracted from type strains` genomes of genera affiliated with the family Roseobacteraceae (formerly Rhodobacteraceae)."
- Typos and inaccuracies on L359, L396, L414 and L415 have been corrected.